# The Interaction Between Alcohol Misuse and Belongingness on Suicidal Ideation Among Military Personnel

**DOI:** 10.3390/ijerph22020246

**Published:** 2025-02-10

**Authors:** Grisel García-Ramírez, Stephen R. Shamblen, Emily Kaner, Roland S. Moore

**Affiliations:** 1Prevention Research Center, Pacific Institute for Research and Evaluation, 2030 Addison St., Ste. 410, Berkeley, CA 94704-2642, USA; ggarcia-ramirez@pire.org (G.G.-R.); ekaner@pire.org (E.K.); 2Pacific Institute for Research and Evaluation, 401 West Main Street, Suite 2100, Louisville, KY 40202-2928, USA; sshamblen@pire.org

**Keywords:** military population, suicidal ideation, alcohol misuse, belongingness

## Abstract

Previous research suggests a high prevalence of suicidal ideation among military personnel. Suicidal ideation is associated with suicide attempts and death. This study focused on the association between belongingness—a component of the Interpersonal Psychological Theory of Suicide—and alcohol misuse on suicidal ideation among the different categories of military branch and military service status. Using the Military Suicide Research Consortium Common Data Elements database (N = 2516), we conducted linear regression analyses to examine the moderating effect of belongingness and alcohol misuse on the association between military branch and military service status (i.e., Active Duty) on suicidal ideation. Results showed a negative significant association between belongingness and suicidal ideation, and a positive significant association between alcohol and suicidal ideation. The results indicated that alcohol misuse moderated the association between military branch and suicidal ideation, but did not moderate the association between military service status and suicidal ideation. Additionally, the results indicated that belongingness moderated the association between military branch and suicidal ideation and the association between military service status and suicidal ideation. The results highlight the differences across military branches and military service statuses and suggest the importance of developing tailored suicide prevention programs to address the specific needs of each military subpopulation.

## 1. Introduction

Suicide is a significant public health concern and a leading cause of death in the United States among U.S. military personnel. Suicide as defined by the Centers for Disease Control and Prevention (CDC) is death as a result of a self-inflicted injury with the intent to die [1]. The annual suicide reports from the U.S. Department of Defense registered an increase in suicide rates among the Active Component (or active duty service) from 20.3 to 25.1 (per 100,000 Service members) between the years 2015 and 2022 [2]. Within the military, suicide prevalence varies by military service status and military branch. Specifically, within military branches, Marines have the highest suicide rates (34.9 per 100,000 service members), followed by the Army (28.9), the Navy (20.6), and the Air Force (19.7) [2]. Because of the clinical and public health significance of suicide, it is vital to understand the risk factors that increase suicidal behaviors in the military population to aid in prevention efforts.

There are several risk factors for suicidal behaviors, including suicidal ideation [3,4] and alcohol misuse [5,6]. Suicidal ideation includes thoughts of suicide and developing plans for suicidal behaviors [7], and is associated with suicide attempts and death [3,8,9]. Alcohol misuse is linked to an increased risk of suicidal ideation, suicide attempts, and death by suicide [5]. Furthermore, research identifies a sense of belongingness as a protective factor against suicidal ideation and behaviors among military personnel [10,11]. While some research focuses on belongingness as a protective factor, other research focuses on thwarted belongingness or a lack of belongingness as a risk factor for suicidal ideation and behaviors [12]. Given the increasing rates of suicide among military service members [2,13], previous research highlights the importance of distinguishing between risks for suicidal ideation and suicide attempts [14], and understanding potential pathways to suicidal behaviors [15,16]. While it remains a challenge to predict the transition from suicidal ideation to suicide attempts, Naifeh and colleagues [14] identified how a history of self-injurious thoughts and behaviors may be a characteristic of soldiers at risk of attempting suicide.

Previous research, important datasets of military personnel (e.g., the Health Related Behaviors Survey (HRBS) [17,18]) and reports by the Department of Defense [2] provide information on alcohol misuse and suicide prevalence among different military branches. Additionally, alcohol misuse and thwarted belongingness are associated with higher risk for suicidal ideation and suicidal behaviors. However, less in known about how these risk factors—alcohol misuse and thwarted belongingness—may impact the association between military status and suicidal ideation. Therefore, in this study we aim to explore the moderating effects of alcohol misuse and belongingness and the association between military branch, military service status, and suicidal ideation.

### 1.1. Alcohol Misuse and Suicidality Among the Military

Alcohol misuse is associated with suicidal ideation and suicide risk among the military population [17,19]. A meta-analysis by Moradi, Dowran, and Sepandi [20] suggested a 9% prevalence of suicidal ideation among military members who consumed alcohol (95% CI: 4–13%), and an 11% prevalence of suicidal ideation in the military (95% CI: 10–13%). Importantly, and based on the Health Related Behaviors Survey (HRBS), 35.3% of military personnel met the criteria for possible alcohol use disorder (AUD), and in 2015, 30% of military personnel were binge drinkers, compared to 24.7% of the U.S. general population [17]. Within the different branches of the U.S. military, research suggests the Army has the highest new onset of AUD, followed by the Navy [21]. Alcohol use and misuse is a significant risk factor for suicidal behaviors among military service members [22,23,24]. Additionally, substance use disorder, including alcohol abuse, can contribute to the transition from suicidal ideation to non-lethal suicide attempts [6]. Specifically, AUD increases the risk of suicidal ideation, suicide attempts, and completed suicide [5]. Previous research suggests service members who have been drinking are likely to act faster on their suicidal impulses, thus increasing their ability to act on suicidal ideation. Furthermore, previous research highlights that drinking to “fit in” or to belong is associated with suicidality among U.S. Army members [9].

### 1.2. The Interpersonal Psychological Theory of Suicide

To understand the mechanisms underlying suicide risk and behaviors, the Interpersonal Psychological Theory of Suicide hypothesizes that suicide risks and suicidal behaviors result in part from the simultaneous presence of thwarted belongingness, perceived burdensomeness, and the ability to act on suicidal ideation [12,25,26]. Thwarted belongingness refers to feelings of loneliness, disconnection from others, and a perceived lack of support from others [26]. Perceived burdensomeness is the belief an individual has that they are a burden to others and that other people would benefit from their death [26]. Yet these two cognitions are not sufficient to act on suicidal impulses. When the ability to act on suicidal impulses is added to thwarted belongingness and perceived burdensomeness, then a person might be at risk of suicidal behaviors [25,26]. Research provides evidence for thwarted belongingness and perceived burdensomeness as mediators of the association between alcohol use and suicide risk [27,28], including in high-risk populations such as firefighters [27]. Thwarted belongingness and alcohol use have also been associated with higher risk of suicidal ideation [29]. While some studies support an association between these constructs, other studies suggest that thwarted belongingness and perceived burdensomeness predict, and are causal mechanisms for, suicidal ideation [30,31,32].

### 1.3. Rationale for the Present Study

It is important to address suicidal ideation before it escalates into suicidal behaviors. The Three-Step Theory (3ST) [33]—the “ideation-to-action” framework—provides an explanation for the progression from suicidal ideation to suicide attempts. Klonsky and May [33] suggest that once suicidal ideation has been developed (Step 1), a person can develop moderate or strong suicidal ideation based on their levels of connectedness (Step 2). Finally, Step 3 suggests three categories of variables that contribute to suicide capacity: (1) dispositional, (2) acquired, and (3) practical. An individual who develops strong suicidal ideation will likely make suicide attempts if they have the capacity [33]. Military personnel might be at higher risk of suicidal behaviors given they might be included in the three categories for suicide capacity in Step 3. For example, military personnel might be desensitized to violence (dispositional), might have experienced pain, injury, and the death of others (acquired), and might have knowledge and access to firearms (practical).

### 1.4. The Present Study

Guided by these theoretical frameworks that seek to better understand the risks and protective factors for suicidal ideation, the current study explores the association between military branch and military service status and suicidal ideation among participants who (a) report alcohol misuse compared to those with no alcohol misuse, and (b) report higher belongingness compared to those with lower belongingness. This study does not establish directional hypotheses, as we explore the association between military branches and military status. Although reports from the Department of Defense [2] provide rates of suicidal behavior by military branch and military service status, instead, we focus upon suicidal ideation; reports about suicidal ideation are not available.

## 2. Materials and Methods

### 2.1. Participants

This study consisted of secondary data analyses of the Military Suicide Research Consortium (MSRC) Common Data Elements (CDE) dataset [34] (N = 2516). The MSRC-CDE dataset consists of aggregated, deidentified, data collected by multiple researchers [35]. Data collection methods varied per site, and included over-the-phone, in-person, and online methods [36]. The inclusion criterion for the current study was that the respondent was currently serving in the military at the time of the survey. This study was approved by the Institutional Review Board of the authors’ research institution and the Department of Defense (DoD) Human Research Protection Office (HRPO).

### 2.2. Measures

#### 2.2.1. Demographic Characteristics

Respondents were asked to report their age, gender, ethnicity, race, military branch (Army, Air Force, Navy, Marine, Coast Guard, or other) and military service status (i.e., Active Duty) at time of service (1 = yes, 0 = no) (see Table 1).

#### 2.2.2. Alcohol Misuse

The Alcohol Use Disorders Identification Test—Consumption (AUDIT-C; [37]) is a 3-item measure to determine the risk of probable AUD, with scores ranging from 0 to 12 (e.g., “How often do you have a drink containing alcohol?”). Cut-off scores of ≥4 for males, and ≥3 for females indicate probable AUD. We converted this variable from a continuous score to a dichotomous score to represent alcohol misuse (0 = no alcohol misuse, 1 = alcohol misuse). A recent study showed that the AUDIT-C has high test–retest reliability for total scores (ICC  =  0.87, 95% CI: 0.87–0.87) [38], and strong validity [37].

#### 2.2.3. Belongingness

The Interpersonal Needs Questionnaire (INQ) is a 15-item measure of thwarted belongingness and perceived burdensomeness [39]. For the MSRC-CDE, 5 items for thwarted belongingness were selected. Respondents rated if the items reflected how they have been feeling recently based on a Likert scale from 1 (not at all true for me) to 7 (very true for me). The sum of these scores ranges from 0 to 35, with higher scores indicating that respondents believed they belonged and felt connected to others. In this study, the Cronbach’s alpha coefficient was strong for belongingness (α = 0.91), similar to previous studies that used the 9-item subscale for thwarted belongingness [40].

#### 2.2.4. Suicidal Ideation

The Depressive Symptom Index-Suicidality Subscale (DSI-SS; [41] is a four-item measure that assesses the frequency and intensity of suicidal ideation, with scores ranging from 0 to 12, as respondents rated items on a 4-point Likert scale. Higher scores reflect greater severity of suicidal ideation at the time participants completed the survey. This scale shows high internal consistency (α = 0.90), and strong convergent validity [42].

### 2.3. Data Analytic Plan

The authors conducted descriptive analyses and multiple ordinary least squares linear regression to examine the associations between alcohol misuse, belongingness, military branch (i.e., Army, Air Force, Navy, and Marines), military service status, and suicidal ideation in Stata, version 18 [43]. Three different models were explored; the first model (Model 1, see Table 2) explored the associations of alcohol misuse, belongingness, military branch, and military service status with suicidal ideation. To assess the moderation effect of alcohol misuse, in the second model (Model 2, see Table 2), the interaction terms alcohol misuse × military branch and alcohol misuse × military service status were included. To assess the moderation effect of belongingness, in the last model (Model 3, see Table 2), the interaction terms belongingness × military branch and belongingness × military service status were included. For all linear regression models, we included demographic characteristics (age, gender, race, and ethnicity) as covariates.

## 3. Results

### 3.1. Descriptive Analyses

Most respondents belonged to the Army (62.31%) and the Navy (22.38%), and 85.36% were on Active Duty. The mean response for suicidal ideation was relatively low (M = 2.36, SD = 3.04), the mean for belongingness was relatively high (M = 23.96, SD = 8.59), and 34.38% fell under the category of alcohol misuse. All other demographic characteristics can be found in Table 1.

### 3.2. Regression Analyses for Suicidal Ideation

Linear regression results for our first model (see Model 1 in Table 2) suggested a significant positive association between alcohol misuse (*β* = 0.08, *t* = 4.04, *p* < 0.001) and suicidal ideation, and a significant negative association between belongingness (*β* = −0.40, *t* = −20.04, *p* < 0.001) and suicidal ideation. The results also showed differences between military branches, suggesting higher suicidal ideation for Navy (*β* = 0.10, *t* = 4.68, *p* < 0.001) and Marines (*β* = 0.10, *t* = 5.00, *p* < 0.001) members than Army members. Finally, respondents whose military service status was Active Duty (*β* = 0.13, *t* = 6.43, *p* < 0.001) had a higher association with suicidal ideation than those who were not in Active Duty status at their time of service.

#### 3.2.1. Alcohol Misuse as Moderator

Linear regression results indicated a significant interaction between alcohol misuse and military branch with suicidal ideation, suggesting a difference for suicidal ideation between Army members and Marines (*β* = −0.05, *t* = −1.97, *p* < 0.001) and between Army and Navy members (*β* = −0.07, *t* = −2.50, *p* < 0.001) (see Model 2 in Table 2). The standardized beta indicated a small effect size for the association between alcohol misuse and suicidal ideation for Marines compared to Army members, and Navy members compared to Army members. A test of simple slopes revealed a negative association between alcohol misuse and suicidal ideation with military branch among participants who belonged to the Navy (*B* = −0.76, *p* = 0.012) and Marines (*B* = −0.69, *p* = 0.049) compared to Army members.

The results did not indicate a significant interaction between alcohol misuse and Active Duty (*β* = 0.06, *t* = 0.90, *p* = 0.367) and suicidal ideation. Main effects did not show a significant positive association between alcohol misuse and suicidal ideation (*β* = 0.07, *t* = 1.23, *p* = 0.220), but showed a significant negative association between belongingness and suicidal ideation (*β* = −0.40, *t* = −20.05, *p* < 0.001). Additionally, the main effects of military branch showed a significant difference between Army members and Navy members (*β* = 0.14, *t* = 5.16, *p* < 0.001), and between Army members and Marines (*β* = 0.13, *t* = 5.12, *p* < 0.001), suggesting that Army members had lower rates of suicidal ideation than Navy members and Marines. Finally, respondents whose military service status was Active Duty (*β* = 0.12, *t* = 5.09, *p* < 0.001) had a higher association with suicidal ideation than those who were not in Active Duty at their time of service. The standardized beta indicated a small effect size for military branch and military service status, and a moderate effect size for belongingness on suicidal ideation.

#### 3.2.2. Belongingness as Moderator

The results of the linear regression indicated a significant interaction between belongingness and military branch with suicidal ideation, suggesting a difference for suicidal ideation between Army and Marines members (*β* = 0.31, *t* = 5.73, *p* < 0.001), but not between Army and Navy members (*β* = 0.004, *t* = 0.06, *p* < 0.950), (see Model 3 in Table 2). The standardized beta indicated a moderate effect size for the association between belongingness and suicidal ideation for Marines compared to Army members, whereas the standardized beta indicated a small effect size for the association between belongingness and suicidal ideation for Navy compared to Army members. A test of simple slope revealed that there was a positive association between belongingness and suicidal ideation among Army members compared to Marines (*B* = 0.13, *p* < 0.001).

Results indicated a significant interaction between belongingness and Active Duty (*β* = −0.42, *t* = −4.77, *p* < 0.001). The standardized beta indicated a moderate effect size for the association between belongingness and suicidal ideation for those whose service status was Active Duty. A test of simple slope revealed a negative association between belongingness and suicidal ideation among participants who were in Active Duty (*B* = −0.11, *p* < 0.001) compared to those who were not in Active Duty at their time of service. Additionally, the main effects showed a positive significant association between alcohol misuse and suicidal ideation (*β* = 0.07, *t* = 3.83, *p* < 0.001), and a negative significant association between belongingness and suicidal ideation (*β* = −0.16, *t* = −2.67, *p* < 0.01). Additionally, the main effects of military branch showed a significant difference between Army members and Marines (*β* = −0.20, *t* = −3.54, *p* < 0.001), suggesting that Army members had a higher association with suicidal ideation than Marines. Finally, there was a significant main effect for Active Duty (*β* = 0.49, *t* = 6.22, *p* < 0.001), suggesting that participants who were in Active Duty at their time of service had a higher association with suicidal ideation than those who were not in Active Duty status at their time of service. The standardized beta indicated a small effect size on suicidal ideation for alcohol misuse, belongingness, and military branch, and a moderate effect size for whose service status was Active Duty.

## 4. Discussion

Suicidal ideation is an important risk factor for suicidal behaviors among the military population. The present study explored the association between military branch and suicidal ideation and military service and suicidal ideation among participants who reported alcohol misuse compared to those with no alcohol misuse. This study also explored the association between military branch and suicidal ideation and military service and suicidal ideation among participants who reported a higher sense of belongingness compared to those with a lower sense of belongingness. Thus, we explored if alcohol misuse and belongingness moderated the association between military branch and military service status, and suicidal ideation. The results indicated that alcohol misuse moderated the association between military branch and suicidal ideation but did not moderate the association between military service status and suicidal ideation. Additionally, results indicated that belongingness moderated the association between military branch and suicidal ideation and the association between military service status and suicidal ideation. This study supports the importance of alcohol misuse and belongingness as contributors to suicidal ideation [44].

Our results are consistent with previous studies indicating that military service members with alcohol misuse are more likely to report suicidal ideation [45,46]. This was particularly true, as results indicated that alcohol misuse moderated the association between suicidal ideation and military branch. Thus, Army members reported less suicidal ideation than Navy members and Marines who also reported alcohol misuse. These results support previous studies that indicate that heavy episodic drinking and AUD are higher in the Marine Corps and Navy than for Army members [47], thus highlighting the importance of alcohol use prevention and AUD treatment among the military population. However, the results did not indicate that alcohol misuse moderated the association between military service status and suicidal ideation. These results indicate a variability in the impact of alcohol misuse on suicidal ideation based on the branch of the military the person belongs to, but not on their military service status. This information can inform the development of alcohol use interventions that can address each specific military branch culture and other possible alcohol misuse risk factors specific to each military branch.

The results also indicated that belongingness moderated the association between military branch and suicidal ideation, and military service status and suicidal ideation. The results indicated that Army members who had a higher sense of belongingness had lower rates of suicidal ideation compared to Marines. Additionally, results indicated that respondents who were in Active Duty reported higher suicidal ideation when reporting a lower sense of belongingness compared to those who were not in Active Duty status at their time of response and also had a lower sense of belongingness. Naifeh et al. [48] suggest that different life contexts and deployment experiences, including combat exposure, between Active Duty service members likely lead to differential processing of traumatic risk factors. These experiences could lead to depression and PTSD, which in turn are associated with suicidal ideation [49]. These fundamental differences in service may offer a reason why Active Duty respondents and different military branches were more likely to experience suicidal ideation. Similarly to the previous results with alcohol misuse as a moderator, these results indicate variability in the impact of belongingness on suicidal ideation based on the military branch a person belongs to and whether they were on Active Duty. This information can inform the development of interventions to enhance a sense of belongingness that can address each specific military branch culture and other possible suicidal risk factors specific to each military branch and military service status.

Our results suggest that a strong sense of belongingness is a protective factor that can buffer against suicidal ideation. Results indicate that respondents who have a stronger sense of belongingness (i.e., less disconnection from others and perceived lack of support from others) are less likely to report suicidal ideation. The results support previous studies on the Interpersonal Psychological Theory of Suicide, and specifically the construct of belongingness, suggesting that those with lower levels of belongingness are more likely to experience suicidal ideation [50]. This underscores the importance of exploring and creating belongingness within the military population, especially since previous studies have suggested that reducing thwarted belongingness among veterans reduced their suicide risks [32].

### 4.1. Limitations and Future Studies

This study includes some limitations. First, although this is a large data set, the MSRC-CDE sample might not be representative of all the national military and currently serving members. This is reflected by the differences in group sizes, particularly among military branches. Another limitation is the self-report nature of the questionnaire, in which respondents might answer items relating to alcohol use and suicidal ideation in a socially desirable way, with recall errors. Furthermore, this dataset did not include information about sexual orientation, rank, or years of military service of respondents. Although there is a relatively high prevalence of AUD among military personnel, we used the AUDIT-C measurement as an indicator of alcohol misuse, rather than a measure of AUD. Given that the Department of Defense does not diagnose an AUD solely on the basis of the AUDIT-C, its findings are not clinically diagnostic of alcohol problems [51]. Finally, the cross-sectional nature of the data does not allow for establishing temporal relationships between alcohol misuse and suicidal ideation, and precludes causal interpretations of the findings.

To improve our understanding, future studies could analyze acute alcohol use, and other substance use disorders including cannabis (CUDs) and opioids (OUDs) [52,53] and their associations with suicidal ideation among military personnel. Longitudinal studies would be beneficial in determining causality of suicidal ideation and suicidal behaviors. For example, future studies could focus on the temporal relationships between deployment, alcohol consumption or substance use disorders, and suicidal ideation among military personnel. Conducting interviews per military branch, via qualitative exploration of these relationships in their different contexts, would be worth pursuing to learn more nuanced information about risk and protective factors for different military branches and military service statuses.

### 4.2. Prevention Implications

Screening for multiple suicide risk factors is important for continuum of care in prevention and treatment to reduce the likelihood of suicidal behaviors. Recent prevention programs have become more comprehensive; for example, including bystander intervention training, reducing alcohol use to reduce suicide risk [54], and focusing on cognitive behavioral therapy (CBT) [55]. Similar programs could be created and would be helpful for future suicide prevention programs by addressing alcohol, other drug use, and poly-use. To reduce alcohol misuse, studies have suggested carefully studying the alcohol drinking norms of each military branch to best address the alcohol use injunctive norms of the branches and peer groups, and to tailor interventions as needed [56]. Other studies on belongingness and the military have suggested the development of technology-based interventions given that lack of belongingness based on personal technology usage was associated with higher suicidal ideation [57]. An integrated intervention addressing suicidal ideation, alcohol use norms, and responsible alcohol use, as well as providing resources to create community and therefore a stronger sense of belongingness, could be beneficial for reducing suicidal ideation. Additionally, it is important for counselors, psychologists, and chaplains to be aware of an individual’s sense of belongingness and provide the appropriate resources and collaborate with military leaders to create opportunities (e.g., community building activities) to enhance a sense of belongingness.

## 5. Conclusions

We found associations between suicidal ideation and alcohol misuse and belongingness, and differences between these associations by military branches and military service statuses. Therefore, a possible implication for this study is the importance of tailoring and adapting different suicide prevention and treatment services to military members based on their military service status and military branch and considering their unique cultures and identities. By tailoring prevention interventions and focusing on military branch, military service status, and specific needs and stressors, suicide prevention treatments may have the capacity to be more effective.

## Figures and Tables

**Table 1 ijerph-22-00246-t001:** Demographic characteristics.

Variable	Total Sample N = 2516	
	n(%)	
Race		
White (%)	1537 (61.78)	
Black/African American (%)	502 (20.18)	
Native American/Alaska Native (%)	28 (1.13)	
Asian (%)	76 (3.05)	
Other (%)	345 (13.87)	
Hispanic (%)	295 (12.51)	
Female (%)	505 (20.07)	
Military branch		
Army (%)	1559 (62.31)	
Air Force (%)	42 (1.68)	
Navy (%)	560 (22.38)	
Marine (%)	341 (13.63)	
Education		
Did Not Complete High School (%)	10 (0.40)	
High School or GED (%)	1110 (44.36)	
Some College (%)	1093 (43.69	
College Degree (%)	221 (8.83)	
More than a College Degree (%)	68 (2.72)	
Active Duty	2116 (85.36)	
Alcohol Misuse (%)	865 (34.38)	
	M (SD)	Value Range
Age	25.64 (6.75)	17–59
Belongingness	23.96 (8.59)	5–35

**Table 2 ijerph-22-00246-t002:** Linear regression for suicidal ideation.

Predictors	Suicidal Ideation								
	Model 1			Model 2			Model 3		
	*B*	SE	95% CI	*B*	SE	95% CI	*B*	SE	95% CI
Age	−0.01	0.01	−0.02, 0.01	−0.01	0.01	−0.02, 0.01	−0.01	0.01	−0.02, 0.01
Race ^1^									
Black/African American	−0.40	0.16	−0.71, −0.10 **	−0.39	0.16	−0.70, −0.08 *	−0.38	0.15	−0.69, −0.08 *
Native American/Alaska Native	0.12	0.53	−0.92, 1.14	0.12	0.53	−0.90, 1.16	0.04	0.52	−0.98, 1.07
Asian	0.25	0.33	−0.41, 0.91	0.26	0.33	−0.39, 0.92	0.15	0.33	−0.50, 0.80
Other	0.09	0.20	−0.30, 0.47	0.09	0.20	−0.29, 0.47	0.09	0.19	−0.29, 0.47
Hispanic	−0.02	0.20	−0.42, 0.37	−0.02	0.20	−0.41, 0.37	−0.09	0.20	−0.48, 0.30
Female	−0.09	0.15	−0.38, 0.20	−0.09	0.15	−0.38, 0.20	−0.07	0.15	−0.36, 0.22
Military Branch ^2^									
Air Force	0.36	0.52	−0.68, 1.39	0.27	0.64	−0.98, 1.54	−1.71	1.81	−5.25, 1.84
Navy	0.71	0.15	0.41, 1.01 ***	1.02	0.20	0.63, 1.40 ***	0.64	0.43	−0.21, 1.48
Marines	0.89	0.18	0.54, 1.23 ***	1.16	0.23	0.71, 1.60 ***	−1.72	0.48	−2.67, −0.76 ***
Active Duty	1.14	0.18	0.80, 1.50 ***	−1.93	0.19	−2.30, −1.55 ***	4.27	0.69	2.94, 5.65 ***
Alcohol Misuse	0.49	0.12	0.25, 0.73 ***	0.45	0.37	−0.27, 1.17	0.46	0.12	0.22, 0.70 ***
Belongingness	−0.14	0.01	−0.16, −0.13 ***	−0.14	0.01	−0.16, −0.13 ***	−0.06	0.02	−0.10, −0.02 **
Alcohol Misuse × Military Branch ^2^									
Air Force	-			0.26	1.13	−1.96, 2.47	-		
Navy	-			−0.76	0.30	−1.35, −0.16 *	-		
Marines	-			−0.69	0.35	−1.38, −0.002 *	-		
Alcohol Misuse × Active Duty	-			0.36	0.40	−0.38, 0.20	-		
Belongingness × Military Branch ^2^									
Air Force	-			-			0.08	0.07	−0.05, 0.22
Navy	-			-			0.001	0.02	−0.03, 0.04
Marines	-			-			0.13	0.02	0.08, 0.17 ***
Belongingness × Active Duty	-			-			−0.11	0.02	−0.16, −0.07 ***

Notes: * *p* ≤ 0.05, ** *p* ≤ 0.01, *** *p* ≤ 0.001; ^1^ White is the referent group. ^2^ Army is the referent group. Model 1: R^2^ = 0.27, F(13, 2073) = 59.77, *p* < 0.001. Model 2: R^2^ = 0.28, F(17, 2069) = 46.29, *p* < 0.001. Model 3: R^2^ = 0.29, F(17, 2069) = 49.89, *p* < 0.001.

## Data Availability

The data that support the findings of this study are available by request to the Military Suicide Research Consortium (MSRC). The authors are not authorized to share the dataset.

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
