# Peer review of "The Interaction Between Alcohol Misuse and Belongingness on Suicidal Ideation Among Military Personnel"

_ijerph, 2025, doi:10.3390/ijerph22020246_

Round 1
Reviewer 1 Report
Comments and Suggestions for Authors
The manuscript is highly relevant and methodologically sound. It has the potential to significantly improve our understanding of suicide risk factors in military populations. The writing is clear, but minor editorial adjustments could improve readability. See my more specific comments for details.
The manuscript frequently uses passive voice, such as “Research has provided evidence…” or “Previous research has also shown…” Authors should remove passive voices where they appear. Additionally, the authors switch between past and present tense in reference to the study. Authors need to select a voice and stick with it throughout the manuscript (for example see lines 117-121).
Abstract:
While the abstract succinctly outlines the study, authors should consider clarifying the role of moderation explicitly to better connect with the methodological details. Strengthening the abstract to better reflect the study's analytical depth and implications would also make it more robust.
Lines 17-18: Was the negative association significant? If it is significant, you should indicate it as such. For example: “belongingness was significantly negatively associated with suicidal ideation.”
Lines 18-19: See previous comment.
Introduction:
Military-specific language should be defined so non-military affiliated, or knowledgeable readers can better comprehend the work. For example, “Active Component” is not a universal term.
A definition of suicide should be provided in the first paragraph. It is important to define terms.
Suicidal ideation should also be defined.
Using statistics is helpful, but the section could benefit from more synthesis of how prior research gaps led to this study.
Lines 36: The statistic about “Active Component” was already mentioned so it is redundant to mention it again. Authors should include this statistic in only one place.
Line 42: References #4 and #5 refer to alcohol use disorders or alcohol abuse. Therefore, the language in this sentence should match. Alcohol use is very different from misuse or abuse of alcohol. Alcohol misuse is mentioned later in line 44. I suggest that the authors make this correction throughout the manuscript.
Line 53: Again, is it alcohol use or misuse? In which case, that could change the information and references you use in the paragraph titled “Alcohol Use and Suicidality Among the Military.” I suggest the authors determine if they would like to focus on alcohol misuse or use, and revise the paragraph accordingly.
Lines 74-76: This sentence seems redundant as this information was previously stated right beforehand.
Materials & Methods:
The reliability and validity of the AUDIT-C, INQ, and DSI-SS should be mentioned.
Dichotomizing continuous variables like AUDIT-C into binary categories can reduce variability and predictive power. Consider justifying this choice or revisiting the raw scale for richer insights.
It is okay to use “we” within manuscripts, however, authors must then choose between past and present tense language as previously indicated.
Lines 155-167: The models are mentioned, but their locations in Table 2 are not. For readers to follow better, the authors should include (Model 1, see Table 2), and so on.
Results:
The analysis is appropriate and well-detailed, especially with interaction terms to assess moderation. Table 2 is highly organized and detailed.
Ensure consistency when presenting results; some effect sizes (e.g., for Navy vs. Marines) could benefit from clearer explanations for their practical significance.
The p-values and t’s should be italicized when reporting statistics. For example, “Results also indicated 212 a significant interaction between belongingness and Active Duty (β = -.42, t = -4.77, p < 001).”
Discussion:
The emphasis on tailored interventions for military branches is valuable, but further elaboration on how such interventions could be operationalized would enhance the discussion.
Line 242: Both the terms results and findings are used concurrently in this sentence. It seems like authors may have intended to only use one term as only one is necessary.
Lines 250-253: What do these results indicate? Why are they relevant? After each result, authors should consider explaining the relevance before moving on to the next result.
Limitations:
Line 281: What does it mean by “inpatient”? Are they currently receiving treatment for suicidal ideation?
Lines 285-288: I am unsure why the authors indicate this is a limitation. Are you trying to say that many military personnel report AUD which could have thwarted alcohol misuse results? The AUDIT is a reputable mechanism for observing alcohol misuse and abuse.
Lines 291-293: What would authors suggest incorporating other substances? Does research indicate that other substance abuse is prevalent? Could it have swayed the results in this study? It is unclear why this is a relevant limitation.
Lines 293-297: It seems like this information should come directly after Lines 288-290.
Implications:
Expanding on how belongingness-building initiatives could be specifically adapted to military contexts would further strengthen this section.
References:
#5 –The hyperlink does not work.
Reviewer 2 Report
Comments and Suggestions for Authors
Dear author,
The manuscript titled "The Interaction Between Alcohol Misuse and Belongingness on Suicidal Ideation Among a Military Population" is a critical research project. Suicide rates among military personnel are high and frequently ignored. The paper, by presenting a significant number of data from military personnel, provides insight into the possible causes of high suicide ideations among this population, demonstrating links to belongingness and alcohol use. The paper's theoretical framework is based on interpersonal theory and the three-step theory of suicide. The work is highly relevant in the military context. I have provided suggestions.
1. Places where the meaning of the sentence is unclear.
2. Inclusion of the importance of portective factor in understanding suicide risk because belongingness is presented here as a protective factor.
3. Minor corrections mentioned in attached file
Best wishes

Reviewer 3 Report
Comments and Suggestions for Authors
The Interaction Between Alcohol Misuse and Belongingness on Suicidal Ideation Among a Military Population
1. ABSTRACT
· Kindly include the setting, pre-testing of the instrument, and the methodology that was used for the study, including the designs.
· Indicate whether the issue of the consent was written or verbal.
2. MATERIALS AND METHODS
· Kindly indicate the approach used and the designs
· Indicate the setting of the test regarding whether it was conducted in a room, online, or outside the room.
· Indicate if the pre-test was done. How many participants were involved in a pre-test? Were the findings of the pre-test included in the main study or not?
· Indicate whether the issue of the consent was a written or a verbal one.
3. RESULTS
· Indicate the reasons for choosing multiple regression analysis instead of other analyses like Chi-square.
4. REFERENCES
· Some references are older than ten years. Kindly use the recent references.
